# Total and Leached Arsenic, Mercury and Antimony in the Mining Waste Dumping Area of Abbadia San Salvatore (Mt. Amiata, Central Italy)

Federica Meloni [1,*], Giordano Montegrossi [1,2], Marta Lazzaroni [1,2,3], Daniele Rappuoli [4,5], Barbara Nisi [1,2] and Orlando Vaselli [1,2,3,*]

1   INSTM—National Interuniversity Consortium of Materials Science and Technology, Via Giusti 9, 50121 Florence, Italy; giordano.montegrossi@igg.cnr.it (G.M.); marta.lazzaroni@unifi.it (M.L.); barbara.nisi@igg.cnr.it (B.N.)
2   CNR-IGG, Institute of Geosciences and Earth Resources, Via G. la Pira 4, 50121 Florence, Italy
3   Department of Earth Sciences, Universita degli Studi Firenze, Via G. la Pira 4, 50121 Florence, Italy
4   Unione dei Comuni Amiata Val d'Orcia, Unità di Bonifica, Via Grossetana 209, Piancastagnaio, 53025 Siena, Italy; d.rappuoli@uc-amiatavaldorcia.si.it
5   Parco Museo Minerario di Abbadia San Salvatore, Via Suor Gemma, Abbadia San Salvatore 1, 53021 Siena, Italy
*   Correspondence: chiccafede95@gmail.com (F.M.); orlando.vaselli@unifi.it (O.V.)

**Abstract:** Total and leached Arsenic, Mercury and Antimony were determined in the topsoils developed on the mining waste dumping area of Le Lame (Mt. Amiata, central Italy) where the post-processing Hg-rich ore deposits from the mining area of Abbadia San Salvatore were stored. The concentrations of As, Hg and Sb were up to 610, 1910 and 1610 mg kg$^{-1}$, respectively, while those in the leachates (carried out with $CO_2$-saturated MilliQ water to simulate the meteoric water conditions) were up to 102, 7 and 661 μg·L$^{-1}$, respectively. Most aqueous solutions were characterized by Hg content <0.1 μg·L$^{-1}$. This is likely suggesting that the mine wastes (locally named "*rosticci*") were possibly resulting from an efficient roasting process that favored either the removal or inertization of Hg operated by the Gould furnaces and located in the southern sector of Le Lame. The highest values of total and leachate mercury were indeed mostly found in the northern portion where the "*rosticci*", derived by the less efficient and older Spirek-Cermak furnaces, was accumulated. The saturation index was positive for the great majority of leachate samples in Fe-oxy-hydroxides, e.g., ferrihydrite, hematite, magnetite, goethite, and Al-hydroxides (boehmite and gibbsite). On the other hand, As- and Hg-compounds were shown to be systematically undersaturated, whereas oversaturation in tripuhyte ($FeSbO_4$) and romeite ($Ca_2Sb_2O_7$) was evidenced. The Eh-pH diagrams for the three chalcophile elements were also constructed and computed and updated according to the recent literature data.

**Keywords:** central Italy; Mt. Amiata; mining waste dumps; mercury; topsoils; leachates

## 1. Introduction

The worldwide sources of heavy elements in surface environments mostly originate in mining activities such as mine exploitation, ore processing and waste disposal in landfills [1]. According to [2,3], tailings accumulated during ore processing have often been stored in steep piles prone to erosion, thus becoming a potential pollution source for the surrounding environment. The impact of mine waste on the environment can pose serious risks since tit can contain reactive mineralogical phases able to generate acidic drainage (e.g., [4,5]) and release toxic elements to the surface and groundwater systems [6,7].

In contaminated soils, heavy metals can be mobilized and transferred to surface and ground waters and absorbed by soil biota [8,9]. According to [1], mobilization of heavy metals is affected by several factors, e.g., redox potential, pH, and bacterial activity.

Specifically, in soils heavy metals can be: (1) associated with water-soluble phases; (2) adsorbed on the surface of solid phases (e.g., clay minerals, organic compounds); (3) related to acid-soluble solid phases (e.g., carbonates) by precipitation and/or co-precipitation; (4) fixed to the surface of reducible solid phases, such as Fe-oxy-hydroxides; (5) bound to insoluble organic matter, e.g., humic acids, or precipitated and/or co-precipitated such as sulfides and (6) included in the crystalline lattice of minerals, stable in soils as residual and newly formed minerals.

The Mt. Amiata district (central Italy) has been classified as the fourth largest producing Hg-district ([10], and references therein). According to [11], in 1925 about 50% of mercury worldwide was provided by the Mt. Amiata mining areas, and about 70% of it was from the Abbadia San Salvatore (ASS) mine (Figure 1).

Several presently-inactive mines are distributed in the Mt. Amiata district [12,13] from where cinnabar(HgS)-rich ore deposits were exploited, although they also contained As-Sb sulfides [1,14]. The origin of Hg mineralization in this area was interpreted as being due to shallow hydrothermal convective systems related to geothermal anomalies that, in turn, were associated with the emplacement of granitoid rocks in the mid- and upper-crustal levels of southern Tuscany during the Pliocene-Pleistocene [15,16]. The ascent of hydrothermal fluids and the consequent emplacement of hydrothermal breccia bodies also allowed the formation of Hg-rich ore deposits in argillitic and calcareous levels [1]. The circulating hydrothermal fluids favored the precipitation of cinnabar and Sb- and As-(e.g., realgar: $As_4S_4$)-sulfides. The Hg-rich ore deposits were exploited from the nineteenth century until about forty years ago [17,18].

In this paper, we present and discuss the geochemical results obtained from the topsoils developed on the waste material from Le Lame mining landfill. In addition to mercury (Hg), arsenic (As) and antimony (Sb) were also considered because these heavy metals are usually associated with Hg-mineralized areas in Tuscany [18]. The main goals of this study were aimed at: (1) determining the total amount of Hg, As and Sb in 35 waste topsoils; and (2) quantifying the release of Hg, As, and Sb from leaching tests carried out with $CO_2$-saturated Milli-Q water, and comparing the data with those reported by the Italian Legislative Decree 152/2006 for drinking waters; (3) assessing the speciation of As, Hg, Sb and (4) defining whether the three chalcophile elements were preferentially partitioned in solution or were co-precipitating with other minerals. Such simulations were carried out by the PHREEQC geochemical code.

## 1.1. The Abbadia San Salvatore Mining Area

The ASS mining area is located on the eastern flank of the 200–300 ky old volcanic silicic complex of Mt. Amiata. The composition of the volcanic products is trachytic to olivine latitic and mostly consisting of lava flows and domes [19–22]. According to ([23] and reference therein), the 65 ha wide ASS mining area was divided into 7 different sectors (Figure 1), each one being characterized by different criticalities in terms of Hg concentration. While sectors 0 and 1 represent areas where Hg was found at low concentrations, sectors 2 and 3 host buildings of miners and management executives, the grounding area of the mining material, the conveyors and the Garibaldi well and old furnaces, dryers, and condensers, respectively. Sector 4, namely "Le Lame", is the site where most mining wastes were accumulated. Sector 5 includes the armory and the house of the guardians. Eventually, Sector 6 became the most Hg-contaminated area, since this portion hosted the edifices of the Gould and Nesa furnaces, the new driers and condensers, and the main mining material storage area.

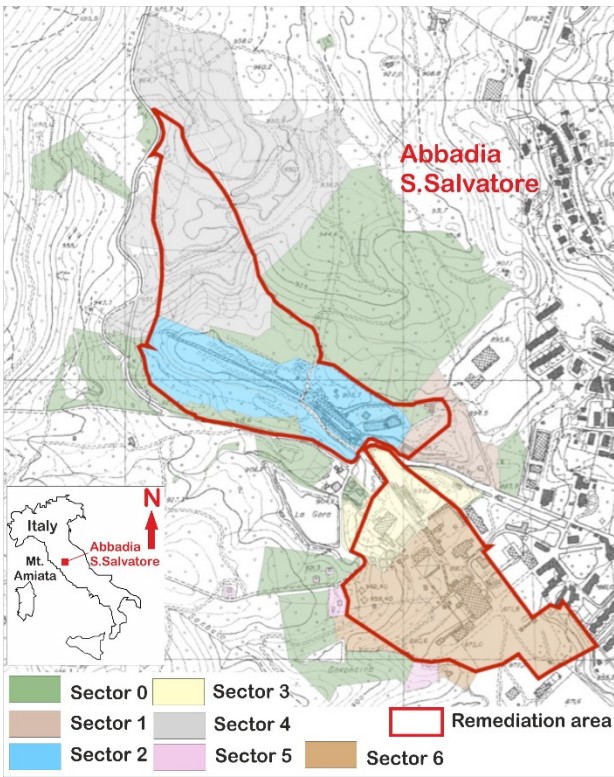

**Figure 1.** Location of the Hg-mining district of Abbadia San Salvatore (Central Italy) and the seven sectors into which the mining are was divided according to the different reclamation activities (modified from [24]). Scale 1:15,000.

Between 1897 and 1909, the two largest Hg-deposits from the Mt. Amiata district were discovered at Abbadia San Salvatore. In [25] reported that the exploited ore had a content between 0.6 and 2.0 wt% of Hg and was exploited up to the depth of 400 m [26].

Liquid mercury was obtained from retort or rotary furnaces (e.g., Spirek, Cermak-Spirek and Gould furnaces). The cinnabar, inside the furnaces, reacted with $O_2$ up to the temperature of 600–700 °C to produce sulfur dioxide and elemental gaseous mercury, according to the following reaction (1):

$$HgS + O_2 = Hg^0 + SO_2 \tag{1}$$

Sulfur dioxide was removed by water washing, whereas mercury vapors were then condensed by cooling systems to obtain liquid mercury and sold to the international market. The post-roasting products and unprocessed materials were stored in the adjacent building hosting the Spirek and Cermak-Spirek furnaces, which were later destroyed and replaced by the most efficient Gould furnaces. The stored waste material was mainly derived by the ASS area, although tailings from nearby Hg-exploitation sites of Mt. Amiata (e.g., Siele, Bagnore and Morone), whose ganga composition was different from that of Abbadia San Salvatore ([27], and reference therein), were also accumulated. From mid-1950, the mining waste of ASS was stored in silos and then transferred to Le Lame mining dump, which is located in the northern part of the mine concession (Sector 4, Figure 1).

The Study Area

The Le Lame mining landfill (Figure 2) is situated on the eastern flank of Mt. Amiata a few kms north of the Abbadia San Salvatore mine and has an extension of about 120,000 m$^2$ [28]. The transfer of the post-processing products (calcines) (locally named "*rosticci*"), to the mining dump began in the 1950s when, to reach higher production of liquid mercury, new automated and conveyor belt systems to transfer the Hg-rich material

from the extraction well to the Gould Furnaces were established [29]. To the best of our knowledge, no documentation is available on how the material deriving from the roasting plants was distributed at Le Lame. This implies that the waste material deriving from the Abbadia San Salvatore and other Mt. Amiata mining sites cannot be located. Additionally, the slopes of Le Lame were modified in time to avoid slumping or landslide processes. It then became difficult to define the thickness and calculate the volume occupied by the post-roasting products. According to the local miners [30], some indications were obtained concerning the location of the stored waste material deriving from the Gould and Cermak-Spirek furnaces operating at Abbadia San Salvatore and reported in Figure 2 with yellow and pale-green dashed lines, respectively. Previous investigations conducted by [28] on three samples collected from Le Lame dumping area showed that Hg were up to 1500 $\mu g \cdot g^{-1}$, likely due to the presence of undecomposed cinnabar as well as by-product compounds such as metacinnabar, metallic $Hg^0$ and Hg salts formed during cinnabar roasting ([26], and references therein). By a hydrogeological point of view, there was no evidence of springs or small creeks, likely due to the fact that the soils and the waste materials and the underlying rocks are rather permeable and the meteoric waters tend to infiltrate and, possibly, mix with the volcanic aquifer. This is also supported by previous investigations on the shallow volcanic aquifer [31] that did not report the presence of anomalous concentrations of As, Hg and Sb.

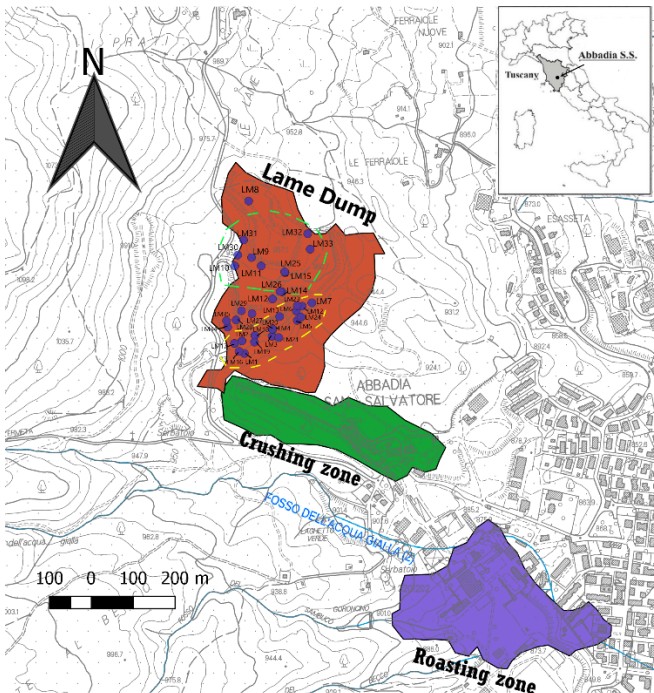

**Figure 2.** The study area of Le Lame (in orange) and the crushing (in green) and roasting (in purple) zones (modified after [1]). The purple dots refer to the soil samples collected for the present study. The yellow and pale-green dashed lines are probably defining the areas where the post-roasting processes by the Gould and Cermak-Spirek furnace were located, respectively.

## 2. Materials and Methods

In August 2017, 35 topsoil (0–30 cm) samples developed on the Hg-processed material were collected from Le Lame. The geographical coordinates of the sampling sites are listed in Table 1. Each sample was identified with the LM prefix and a progressive number. From each sampling site, at least 3 kg of soils were collected, mixed in a basin and sieved at 2 cm on site. Then, an aliquot of the <2 cm fraction soil was stored into 1 L polyethylene containers and transferred to the laboratory. The soil samples were left drying in the oven at the temperature of 30 °C for about 3 days. Each sample was then sieved at

2 mm following the protocol in accordance with the local environmental protection agency (ARPAT). The Hg, As, and Sb concentrations in soils were measured by ICP-MS (Agilent 7500CE) following the procedures outlined by the U.S. Environmental Protection Agency (EPA): EPA 6020 D (2018). The soil samples were solubilised by acid digestion with the inverse aqua regia method (e.g., $HNO_3$: HCl ratio of 3 to 1, EPA 3051A method) at the Laboratories of C.S.A. Ltd. (Rimini, Italy). Three replicates were performed for each sample. The analytical error was <10%.

**Table 1.** Geographical coordinates of the sampling topsoils.

| Sample | EST | NORD |
|:---:|:---:|:---:|
| LM1 | 717124 | 4751683 |
| LM2 | 717126 | 4751700 |
| LM3 | 717182 | 4751743 |
| LM4 | 717219 | 4751763 |
| LM5 | 717068 | 4752006 |
| LM6 | 717075 | 4751871 |
| LM7 | 717035 | 4751851 |
| LM8 | 717098 | 4751851 |
| LM9 | 717125 | 4751772 |
| LM10 | 717142 | 4751730 |
| LM11 | 717148 | 4751789 |
| LM12 | 717155 | 4751835 |
| LM13 | 717045 | 4751647 |
| LM14 | 717034 | 4751666 |
| LM15 | 717083 | 4751685 |
| LM16 | 717082 | 4751669 |
| LM17 | 717125 | 4751700 |
| LM18 | 717140 | 4751680 |
| LM19 | 717196 | 4751755 |
| LM20 | 717184 | 4751756 |
| LM21 | 717193 | 4751729 |
| LM22 | 717154 | 4751837 |
| LM23 | 717143 | 4751790 |
| LM24 | 717076 | 4751737 |
| LM25 | 717038 | 4751722 |
| LM26 | 717051 | 4751744 |
| LM27 | 717042 | 4751877 |
| LM28 | 717056 | 4751913 |
| LM29 | 717209 | 4751928 |
| LM30 | 717215 | 4751891 |
| LM31 | 717018 | 4751705 |
| LM32 | 717015 | 4751720 |
| LM33 | 717124 | 4751683 |
| LM34 | 717126 | 4751700 |
| LM35 | 717184 | 4751721 |

A leaching test was also carried out on the <2 mm fraction. Specifically, the leaching test consisted in weighing 15 g of topsoil in a beaker with 75 mL of $CO_2$-saturated MilliQ water (top soil/water ratio of 1 to 5) and stirred for 24 h. The $CO_2$-saturated MilliQ water was used to mimic the interaction with meteoric waters [32]. The $CO_2$-saturated MilliQ water was obtained by gently bubbling pure $CO_2$ into a bottle for about 20 min until a pH of 4.5 was reached. The final solution was obtained by filtering (at 0.45 μm) the leachate by using a Sartorius filtering device. The solution was then split into three aliquots. In the first one, the pH was analyzed using a bench-top pH-meter (Crison micro pH 2000) after calibration with a solution at pH = 7 and one at pH = 4 at a temperature of 20 °C. The second one was used for determining the main cations ($Ca^{2+}$, $Na^+$, $K^+$ and $Mg^{2+}$) and anions ($F^-$, $Cl^-$, $NO^-_3$, $PO_4^{3-}$ and $SO_4^-$) by ion chromatography (861 Advanced Compact IC-Metrohm and 761 Compact IC-Metrohm, respectively), while $NH_4^+$

and $HCO_3^-$ were analyzed by colorimetry, according to the Nessler method, by using a molecular spectrophotometer HACH-DR2010, and acidimetric titration with HCl 0.01 N and methyl-orange as indicator, respectively. The main eluate chemistry was carried out at the Department of Earth Sciences in Florence. The analytical error was < 5%. Eventually, the third aliquot was acidified with $HNO_3$ suprapur (1%) (Merck) for the determination for As, Sb, Fe, Al, $SiO_2$ and Mn according to the EPA 6020 B (2014) method. The detection limits were 0.1 $\mu g \cdot L^{-1}$ for As, Mn and Sb, 5 $\mu g \cdot L^{-1}$ for Fe and Al and 0.5 $mg \cdot L^{-1}$ for $SiO_2$. The analytical error was <10% for all these species. Dissolved mercury was determined according to the EPA 7473 (2007) method, using HCl UpA (Romil) to acidify the eluates, at the C.S.A. Ltd. Laboratories in Rimini (Italy). The detection limit was <0.1 $\mu g \cdot L^{-1}$ and the analytical error was <10%. Blanks of $CO_2$-saturated MilliQ waters showed metals and $SiO_2$ contents below the instrumental detection limit, i.e., 0.1 $\mu g \cdot L^{-1}$ for Hg, As, Mn and Sb, 5 $\mu g \cdot L^{-1}$ for Fe and Al and 0.5 $mg \cdot L^{-1}$ for $SiO_2$.

## 2.1. Statistical Analysis

The software R was used for basic statistics to calculate minimum, maximum, mean, median, standard deviation, and skewness for each species, whereas the Anderson-Darling test was utilized to verify the normal distribution of analytical data.

## 2.2. Geochemical Modeling

The pH and total dissolved concentrations of major cations and anions, trace elements (Fe, Al, $SiO_2$ and Mn), Hg, As, and Sb in the eluates were elaborated by the geochemical code PHREEQC version 3.5.0 [33] to estimate the aqueous solution speciation and to construct Eh-pH diagrams. The PHREEQC geochemical code is based on the calculation of equilibrium between aqueous solutions and minerals, gases, solid solutions, exchangers, and sorption surfaces. PHREEQC implements several types of aqueous activity models depending on the applied database; most of them use the Davies equation (B-Dot activity model) as the Lawrence Livermore National Laboratory model [34] and WATEQ4F. The Pitzer specific ion-interaction and the SIT (Specific Ion Interaction Theory) aqueous models are also available in a specific database. In the present study, the thermodynamic properties of some minerals relevant to our simulations were checked and eventually added to the PHREEQC *minteq.v4*, a database derived from MINTEQA2 version 4 [35,36] that uses the B-dot activity model. The thermodynamic data for the reactions added to the *minteq.v4* database are reported in Table 2 with the corresponding references.

**Table 2.** Thermodynamic properties included in the *minteq.v4* database. The solids phases or minerals, the precipitation/dissolution reactions, the Log k and the corresponding reference(s) are reported.

| Solids Phases and Minerals | Reactions | Log k | References |
|---|---|---|---|
| Scorodite | $FeAsO_4 + 3H^+ = Fe^{3+} + H_3AsO_4 + 2H_2O$ | 0.4 | [34] |
| $Ca_5(AsO_4)_3(OH)$ | $Ca_5(AsO_4)_3(OH) = 5Ca^{2+} + 3AsO_4^{3-} + OH^-$ | −40.12 | [37] |
| $Ca_4(OH)_2(AsO_4)_2 \ast 4H_2O$ | $Ca_4(OH)_2(AsO_4)_2 \ast 4H_2O = 4Ca^{2+} + 2AsO_4^{3-} + 2OH^- + 4H_2O$ | −27.49 | [37] |
| $Ca_3(AsO_4)_2 \ast 3H_2O$ | $Ca_3(AsO_4)_2 \ast 3H_2O = 3Ca^{2+} + 2AsO_4^{3-} + 3 H_2O$ | −21.14 | [37] |
| $FeAsO_4$ | $FeAsO_4 = Fe^{3+} + AsO_4^-$ | 13.97 | [38] |
| $FeHAsO_4$ | $FeAsO_4 = Fe^{2+} + HAsO_4^{2-}$ | 3.37 | [38] |
| $MgHAsO_4$ | $MgHAsO_4 = Mg^{2+} + HAs_4^{2-}$ | 2.68 | [38] |
| $CaHAsO_4$ | $CaHAsO4 = Ca^{2+} + HAs_4^{2-}$ | 2.51 | [38] |
| $NaHAsO_4$ | $NaHAsO_4 = Na^+ + H_2AsO_4^-$ | 1.75 | [38] |
| $NaH_2AsO_3$ | $NaH_2AsO_3 = Na^+ + H_2AsO_3^-$ | 0.25 | [38] |
| $KH_2AsO_4$ | $KH_2AsO_4 = K^+ + H_2AsO_4^-$ | 1.89 | [38] |
| $Ca(Sb(OH)_6)_2 \ast 6H_2O$ | $Ca(Sb(OH)_6)_2 \ast 6H_2O = Ca^{2+} + 2Sb(OH)^-_6 + 6H_2O$ | −10.57 | [39] |
| $Ca(Sb(OH)_6)_2$ | $Ca(Sb(OH)_6)_2 = Ca^{2+} + 2Sb(OH)^-_6$ | −12.55 | [40] |
| Romeite | $Ca_2Sb_2O_7 + 5H_2O + 2H^+ = 2Ca^{2+} + 2Sb(OH)^-_6$ | −6.7 | [41] |

**Table 2.** *Cont.*

| Solids Phases and Minerals | Reactions | Log k | References |
|---|---|---|---|
| Tripuhyite | $FeSbO_4 + H_2O + 3H^+ = Fe^{3+} + Sb(OH)^0_5$ | $-10.68$ | [42] |
| Schafarzikite | $FeSb2O_4 + H_2O + 2H^+ = Fe^{2+} + 2Sb(OH)^0_3$ | $-12.21$ | [42] |
| Bystromite | $MgSb_2O_7 + 4H_2O + 2H^+ - = Mg^{2+} + 2Sb(OH)^0_5$ | $-9.44$ | [43] |
| Brizziite | $NaSbO_3 + 3H_2O = Na^+ + Sb(OH)^-_6$ | $-6.70$ | [44] |

The relationships between Hg-Cl-S, As-S, and Sb-S-Ca-Fe were studied through the Pourbaix diagrams (Eh-pH) and calculated according to the point-by-point mass balance method [45]. Following [45], the predominant species at each given point of Eh and pH was identified, taking into account other variables, e.g., temperature, and presence of ligand(s). The predominant species was determined to be the species with the highest contents of the considered element. This predominance involves the absolute element content (in moles) considering both aqueous and solid species. Aqueous phase equilibria were taken into account in the modified *minteq.v4.dat* database file and the solid phases were included through dissolution/precipitation reactions and reported as the logarithm of equilibrium constant (log K) at the standard temperature and pressure condition [46].

## 3. Results

### 3.1. Hg, As and Sb in Waste Topsoils

The concentrations of As, Hg, and Sb at the sampling sites (<2 mm topsoil fraction) and the main descriptive statistics of the studied topsoils (e.g., number of observations, minimum, maximum, mean, median, standard deviation and skewness) are listed in Tables 3 and 4. Note that, in Table 4, unless Hg and As, Sb was only determined in 31 soil samples as four out of 35 had contents <2 mg kg$^{-1}$. The minimum and maximum contents of As, Hg, and Sb in the topsoils were 10, 4 and <2 mg kg$^{-1}$ and 610, 1910 and 1610 mg kg$^{-1}$, respectively. Basically, the concentrations of Hg and Sb were largely variable since they were spanning between three and four orders of magnitude, respectively, while those of As were included within two orders of magnitude. This suggests that the three chalcophilic elements are heterogeneously distributed in the dumping area of Le Lame.

**Table 3.** Concentrations of As, Hg, Sb (mg kg$^{-1}$) in the studied topsoils.

| Sample | As | Hg | Sb |
|---|---|---|---|
| LM1 | 17.0 | 103.0 | <2 |
| LM2 | 43.0 | 34.0 | 14.0 |
| LM3 | 16.0 | 218.0 | <2 |
| LM4 | 97.0 | 60.0 | 338.0 |
| LM5 | 10.0 | 96.0 | <2 |
| LM6 | 36.0 | 551.0 | 7.0 |
| LM7 | 16.0 | 105.0 | 2.0 |
| LM8 | 29.0 | 4.0 | <2 |
| LM9 | 226 | 794.0 | 967.0 |
| LM10 | 25.0 | 112.0 | 4.0 |
| LM11 | 181.0 | 131.0 | 1419.0 |
| LM12 | 25.0 | 31.0 | 4.0 |
| LM13 | 132.0 | 1043.0 | 347.0 |
| LM14 | 23.0 | 18.0 | 3.0 |
| LM15 | 122.0 | 240.0 | 338.0 |
| LM16 | 72.5 | 80.4 | 410.0 |
| LM17 | 90.3 | 13.9 | 62.5 |
| LM18 | 66.1 | 56.8 | 132.0 |
| LM19 | 14.6 | 33.5 | <2 |
| LM20 | 134.0 | 68.0 | 530.0 |
| LM21 | 24.4 | 304.0 | 3.0 |

**Table 3.** *Cont.*

| Sample | As | Hg | Sb |
|---|---|---|---|
| **LM22** | 24.5 | 384.0 | 5.3 |
| **LM23** | 25.5 | 321.0 | 3.4 |
| **LM24** | 15.8 | 124.0 | 6.9 |
| **LM25** | 150.0 | 113.0 | 442.0 |
| **LM26** | 32.1 | 151.0 | 8.8 |
| **LM27** | 33.8 | 18.5 | 7.1 |
| **LM28** | 53.9 | 39.7 | 4.3 |
| **LM29** | 46.7 | 236.0 | 5.3 |
| **LM30** | 124.0 | 38.2 | 162.0 |
| **LM31** | 306.0 | 323.0 | 8.5 |
| **LM32** | 616.0 | 173.0 | 1610.0 |
| **LM33** | 131.0 | 1910.0 | 398.0 |
| **LM34** | 151.0 | 90.1 | 109.0 |
| **LM35** | 131.0 | 206.0 | 15.8 |

**Table 4.** Main descriptive statistics concerning As, Hg and Sb.

| Metals | N.obs. | Min. | Max. | Mean | Median | S.D. | Skewness |
|---|---|---|---|---|---|---|---|
| **As** | 35 | 10 | 616 | 92.6 | 44.85 | 114.5 | 3.14 |
| **Hg** | 35 | 4 | 1910 | 235.0 | 108.5 | 366.3 | 3.37 |
| **Sb** | 31 | 0.18 | 1610 | 237.7 | 14.0 | 408.3 | 2.37 |

*3.2. Leachate Geochemistry*

The pH values and concentrations of the main dissolved species (in $mg \cdot L^{-1}$) determined by leaching tests with $CO_2$-saturated MilliQ water are listed in Table 5. The pH varied from weakly acidic (6.63) to alkaline (8.35). The cation triangular diagram, where the concentrations in $meq \, L^{-1}$ were recalculated to 100%, is reported in Figure 3, demonstrating that $Ca^{2+}$ and $Mg^{2+}$ are the dominant species. This feature is also similar to the groundwaters collected from the mining area of ASS where in the past the roasting products were used to fill a small valley [27]. The anion triangular diagram is not reported, since the use of $CO_2$-saturated MilliQ water artificially increased the $HCO_3^-$ concentration. The $Ca^{2+}$ and $SO_4^{2-}$ concentrations varied from 100 and 270 $mg \cdot L^{-1}$ and 2.7 and 60 $mg \cdot L^{-1}$, respectively. $Na^+$ and $Cl^-$ were always below 7 $mg \cdot L^{-1}$. $Mg^{2+}$, $NH_4^+$, $K^+$ and $NO_3^-$ never exceeded 18, 3.4, 40 and 306 $mg \cdot L^{-1}$, respectively. It is worth mentioning that the highest values of $K^+$ corresponded to those samples where the highest contents of $NO_3^-$ were also found. Finally, $F^-$, $Br^-$ and $PO_4^{3-}$ were detected in a few samples, but never exceeded 1.4, 0.2 and 0.2 $mg \cdot L^{-1}$, respectively.

**Table 5.** pH values and concentrations of main cations and anions ($mg \cdot L^{-1}$) in the $CO_2$-saturated MillQ water eluate samples.

| Sample | pH | $HCO_3^-$ | $F^-$ | $Cl^-$ | $Br^-$ | $NO_3^-$ | $PO_4^{3-}$ | $SO_4^{2-}$ | $Ca^{2+}$ | $Mg^{2+}$ | $Na^+$ | $K^+$ | $NH_4^+$ |
|---|---|---|---|---|---|---|---|---|---|---|---|---|---|
| **LM1** | 6.90 | 544 | | 2.3 | | 22 | | 11 | 191 | 5 | 1.9 | 10 | 0.4 |
| **LM2** | 7.72 | 502 | | 7.0 | | 306 | | 32 | 245 | 18 | 4.0 | 40 | 1.8 |
| **LM3** | 6.63 | 516 | | 1.3 | | 33 | | 10 | 181 | 7 | 1.5 | 15 | 0.5 |
| **LM4** | 6.95 | 420 | | 1.6 | | 45 | | 22 | 151 | 7 | 0.8 | 8 | 0.7 |
| **LM5** | 7.89 | 475 | | 1.0 | | 22 | | 6 | 159 | 7 | 1.4 | 11 | 0.2 |
| **LM6** | 7.20 | 763 | | 0.2 | | 11 | | 19 | 270 | 6 | 0.9 | 6 | 0.4 |
| **LM7** | 7.36 | 420 | | 3.7 | | 10 | | 45 | 157 | 5 | 7.0 | 13 | 0.2 |
| **LM8** | 7.44 | 265 | | 1.7 | | 4 | | 2.7 | 87 | 2.2 | 1.1 | 12 | 0.3 |
| **LM9** | 8.22 | 712 | | 3.2 | | 59 | | 17 | 248 | 12 | 2.0 | 29 | 2.0 |
| **LM10** | 7.46 | 400 | | 2.5 | | 75 | | 10 | 130 | 5 | 1.6 | 19 | 0.8 |
| **LM11** | 7.44 | 716 | | 1.6 | 0.1 | 5 | | 10 | 219 | 9 | 1.1 | 8 | 0.9 |

**Table 5.** *Cont.*

| Sample | pH | HCO$^-_3$ | F$^-$ | Cl$^-$ | Br$^-$ | NO$_3^-$ | PO$_4^{3-}$ | SO$_4^{2-}$ | Ca$^{2+}$ | Mg$^{2+}$ | Na$^+$ | K$^+$ | NH$_4^+$ |
|---|---|---|---|---|---|---|---|---|---|---|---|---|---|
| **LM12** | 6.93 | 673 | | 4.0 | | 116 | | 14 | 237 | 17 | 2.7 | 37 | 3.4 |
| **LM13** | 6.78 | 347 | | 2.2 | | 23 | | 38 | 132 | 6 | 1.7 | 10 | 1.5 |
| **LM14** | 7.09 | 515 | | 4.2 | | 59 | | 32 | 175 | 18 | 5.0 | 10 | 1.7 |
| **LM15** | 8.14 | 495 | | 2.2 | | 11 | | 9.1 | 154 | 13 | 1.6 | 1.8 | 2.3 |
| **LM16** | 7.28 | 575 | | 0.3 | | 21 | | 7.2 | 200 | 6 | 1.1 | 11 | 0.6 |
| **LM17** | 7.08 | 341 | | 1.1 | | 7 | 0.1 | 6.5 | 114 | 3.3 | 1.5 | 3.5 | 0.4 |
| **LM18** | 7.51 | 641 | | 0.5 | | 11 | | 9.5 | 223 | 6 | 1.5 | 9 | 0.5 |
| **LM19** | 7.20 | 451 | | 1.0 | | 15 | | 12.8 | 156 | 6 | 2.2 | 5 | 0.4 |
| **LM20** | 7.77 | 722 | 0.1 | 0.7 | | 29 | | 7.9 | 223 | 11 | 2.1 | 15 | 1.6 |
| **LM21** | 7.29 | 549 | | 0.3 | | 4 | | 7.4 | 181 | 5 | 1.3 | 10 | 0.6 |
| **LM22** | 8.35 | 434 | | 1.4 | | 18 | | 12 | 150 | 5 | 1.4 | 9 | 0.9 |
| **LM23** | 7.17 | 508 | | 2.3 | | 49 | 0.2 | 7 | 175 | 9 | 4.0 | 13 | 2.0 |
| **LM24** | 7.57 | 710 | 0.7 | 1.3 | | 28 | | 8 | 250 | 7.7 | 1.1 | 14 | 0.5 |
| **LM25** | 7.88 | 714 | 0.2 | 1.4 | | 21 | | 10 | 243 | 9 | 2.0 | 13 | 1.1 |
| **LM26** | 7.19 | 658 | | 0.7 | | 5.1 | | 11 | 227 | 5 | 2.9 | 9 | 0.8 |
| **LM27** | 7.45 | 245 | | 1.0 | 0.2 | 9.1 | | 2.8 | 70 | 2.5 | 1.2 | 6 | 2.9 |
| **LM28** | 8.35 | 439 | | 1.1 | | 11 | | 9 | 143 | 2.6 | 4.0 | 9 | 0.4 |
| **LM29** | 7.67 | 597 | 1.4 | 0.5 | | 12 | | 7 | 203 | 7 | 1.2 | 7 | 0.6 |
| **LM30** | 8.28 | 401 | | 0.5 | | 17 | | 4 | 124 | 7 | 0.9 | 6 | 3.0 |
| **LM31** | 7.82 | 286 | | 2.6 | | 7.3 | | 60 | 100 | 4 | 2.9 | 9 | 0.7 |
| **LM32** | 6.71 | 358 | | 0.9 | | 4.5 | | 20 | 125 | 6.2 | 1.2 | 4 | 0.3 |
| **LM33** | 7.08 | 420 | | 2.3 | | 4.3 | 0.1 | 46 | 162 | 4 | 2.5 | 6 | 0.3 |
| **LM34** | 7.84 | 502 | 0.2 | 0.5 | | 11 | | 9 | 155 | 4 | 0.9 | 5 | 1.1 |
| **LM35** | 7.25 | 598 | | 1.3 | | 54 | | 8 | 210 | 7 | 1.6 | 14 | 3.1 |

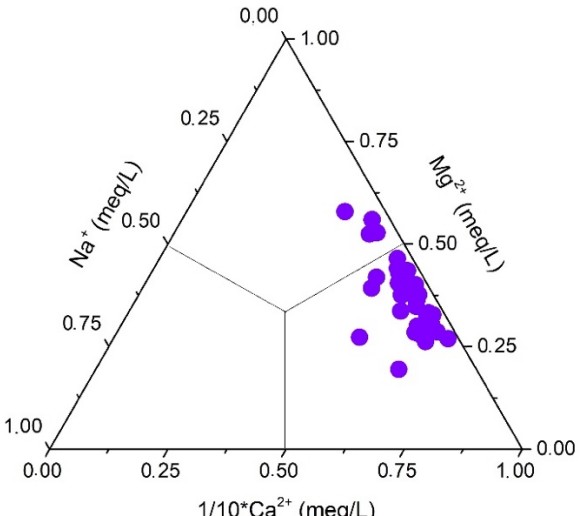

**Figure 3.** The cation triangular diagram of the CO$_2$-saturated water leachates.

The concentrations of metals and metalloids (in µg·L$^{-1}$), ordered according to the increasing atomic number, and SiO$_2$ (in mg·L$^{-1}$) in the leachates, are listed in Table 6. Aluminum, Mn, and Fe varied between 34 and 1148, 3.1 and 280 and 5 and 568 µg·L$^{-1}$, respectively. As and Sb showed concentration up to 102 and 661 µg·L$^{-1}$, respectively. Mercury showed concentrations up to 7 µg·L$^{-1}$ although most samples were below the instrumental detection limit (0.1 µg·L$^{-1}$, Table 4) or slightly higher. Finally, SiO$_2$ contents were included between 3.7 and 10.1 mg·L$^{-1}$.

**Table 6.** Concentrations of metals and metalloids ($\mu g \cdot L^{-1}$) and $SiO_2$ ($mg \cdot L^{-1}$) in the $CO_2$-saturated MilliQ leachates.

| Sample | Al | Mn | Fe | $SiO_2$ | As | Sb | Hg |
|--------|------|-----|-----|--------|------|------|------|
| LM1 | 34 | 108 | 40 | 7.2 | 2.3 | 2 | <0.1 |
| LM2 | 113 | 184 | 45 | 5.8 | 16 | 5 | <0.1 |
| LM3 | 112 | 116 | 56 | 9.5 | 1.6 | 1.9 | <0.1 |
| LM4 | 170 | 47 | 92 | 6.8 | 61 | 27 | <0.1 |
| LM5 | 1148 | 186 | 568 | 9.0 | 3.1 | 2 | <0.1 |
| LM6 | 34 | 280 | 25 | 5.8 | 8.3 | 3.8 | <0.1 |
| LM7 | 42 | 33 | 12 | 7.7 | 3.3 | 2.6 | <0.1 |
| LM8 | 69 | 28 | 21 | 7.6 | <1 | 1.1 | <0.1 |
| LM9 | 173 | 46 | 75 | 9.4 | 102 | 150 | <0.1 |
| LM10 | 80 | 17 | 34 | 5.2 | 11 | 1.4 | <0.1 |
| LM11 | 68 | 65 | 23 | 6.9 | 89 | 119 | <0.1 |
| LM12 | 88 | 204 | 104 | 9.1 | 3.4 | 4.5 | <0.1 |
| LM13 | 117 | 67 | 88 | 5.0 | 40 | 33 | <0.1 |
| LM14 | 55 | 247 | 84 | 6.3 | 2.1 | 2 | <0.1 |
| LM15 | 286 | 31 | 306 | 6.6 | 68 | 39 | <0.1 |
| LM16 | 108 | 82 | 53 | 7.3 | 25.7 | 112 | 0.3 |
| LM17 | 54 | 40 | 23 | 4.5 | 84 | 27 | <0.1 |
| LM18 | 117 | 91 | 50 | 8.4 | 35 | 44 | <0.1 |
| LM19 | 34 | 96 | 56 | 6.7 | 1.7 | 0.8 | 0.2 |
| LM20 | 153 | 61 | 56 | 9.4 | 70.4 | 151 | <0.1 |
| LM21 | 32 | 75 | 19 | 6.7 | 4.9 | 1.2 | <0.1 |
| LM22 | 81 | 69 | 56 | 6.5 | 4.7 | 1.7 | 0.2 |
| LM23 | 171 | 94 | 132 | 9.7 | 0.6 | 1.8 | 7.7 |
| LM24 | 115 | 78 | 45 | 10.1 | 5.2 | 3.2 | <0.1 |
| LM25 | 95 | 42 | 35 | 9.0 | 68 | 106 | 0.1 |
| LM26 | 69 | 158 | 29 | 8.3 | 1.1 | 2.7 | <0.1 |
| LM27 | 191 | 11 | 47 | 4.8 | 7.9 | 1.7 | <0.1 |
| LM28 | 102 | 34 | 109 | 5.9 | 25 | 0.6 | <0.1 |
| LM29 | 109 | 130 | 34 | 8.5 | 26 | 1.5 | <0.1 |
| LM30 | 178 | 57 | 98 | 5.0 | 29 | 60.7 | <0.1 |
| LM31 | 57 | 3.1 | 560 | 4.5 | 0.5 | <0.1 | <0.1 |
| LM32 | 17 | 23 | 5 | 3.7 | 26 | 662 | 0.6 |
| LM33 | 17 | 9.7 | 11 | 5.0 | 35 | 116 | 0.2 |
| LM34 | 225 | 47 | 181 | 6.2 | 50 | 23.8 | <0.1 |
| LM35 | 133 | 101 | 55 | 9.8 | 14 | 10.5 | <0.1 |

*3.3. PHREEQ-C Modeling*

The saturation indices (SI), calculated as the log(IAP)/Ksp, where IAP is the Ionic Activity Product and Ksp the constant of solubility product, of selected minerals computed with the software PHREEQC are reported in the Supplementary Material (Table S1). All samples ended up being saturated in calcite and dolomite, likely due to the effect of using $CO_2$-saturated MilliQ in the leaching tests. Among the silicate phases, kaolinite was systematically saturated, whereas quartz appeared to be in equilibrium with the aqueous solutions. Interestingly, Fe-oxides and Fe-hydroxides, e.g., ferrihydrite, hematite, magnetite, goethite, and Al-hydroxides (boehmite and gibbsite) were always characterized by SI > 0, and the former were apparently controlling the solubility of As and Sb. Nevertheless, Arsenic and Mercury compounds were always undersatured. As far as Sb minerals, tripuhyte ($FeSbO_4$) was always oversaturated, with the exception of LM31 where it was not present. Romeite ($Ca_2Sb_2O_7$) was oversaturated in sample LM9, which showed a pH > 8.

## 4. Discussion

*4.1. The Waste Topsoils*

The concentrations of As, Hg and Sb from Le Lame topsoils, developed on the tailings mainly derived by the former Hg-mining area of Abbadia San Salvatore, showed relatively

high values. By comparing the results of the three elements (Table 3) with those imposed by the Italian Law (Legislative Decree 152/06), 12 and 10 soil samples out of 35 showed that As and Sb had concentrations higher than those for areas intended for industrial use (50 and 30 mg kg$^{-1}$, respectively). Additionally, all samples except LM 8 exceeded the Hg legal limit for areas intended for residential (1 mg kg$^{-1}$) and industrial (5 mg kg$^{-1}$) use.

According to the data in Table 4, the considered variables do not show good symmetry with respect to the arithmetic mean as barycenter, since they are affected by strong positive skewness due to a certain prevalence of low values with a few isolated high values. These observations were confirmed by applying the Anderson Darling normality test where, in each case, a *p*-value << 0.05 was found. According to [47–49], the frequency distribution of elements is rarely showing a normal (Gaussian type) behavior, as they generally have positive skewness approaching a log-normal distribution, i.e., logarithms of concentrations are approximately normally distributed. However, for the sake of a better visualization, the following diagrams are reported in mg kg$^{-1}$. As shown by the violin plots [50] (Figure 4), As and Sb are characterized by two populations, highlighted by the kernel density, as follows: (1) low-to-medium concentrations (10–33.8 and 0.18–62.50 mg kg$^{-1}$, respectively) located in the southern part of the dumping area; and (2) high values (33.8–616, and 63–1610 mg kg$^{-1}$, respectively) in the northern portion.

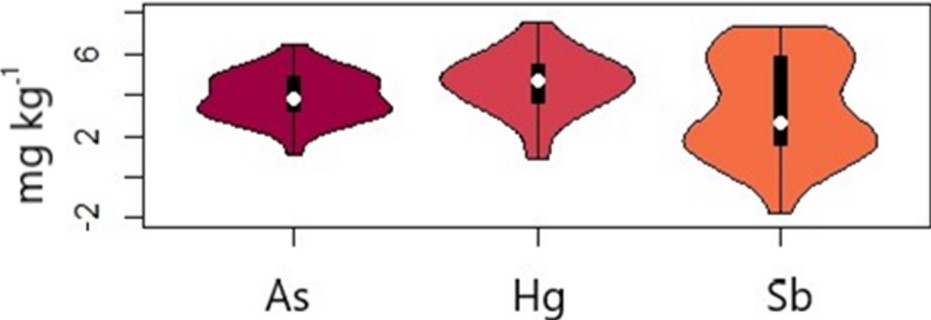

**Figure 4.** As, Hg and Sb violin plots. The *x*-axis shows the three metals, the *y*-axis are the concentrations (mg kg$^{-1}$) expressed as a logarithm.

These two populations also show a different origin of the materials deposited in the La Lame landfill. According to [27], ore and waste materials from other Hg exploitations areas (e.g., Morone, Bagnore and Siele) were different from that of the ASS mine, being generally more enriched in Sb [13,16] and stored adjacent to the building hosting the old Spirek and Cermak-Spirek furnaces. Therefore, it can be assumed that materials from the above-mentioned mines were preferentially accumulated in the northern portion of the Le Lame dumping area. Conversely, a single population characterizes the Hg concentrations, which is mainly related to medium-high values. Additionally, Hg is not correlated with neither As nor Sb (correlation coefficient 0.23) and, consequently, its spatial distribution differs from those of the others' metals, whereas As and Sb have a similar behavior, as also suggested by a significant positive correlation coefficient (0.86). In order to better represent the distribution of these three elements, the dots maps of As, Hg and Sb (in mg kg$^{-1}$) are reported in Figure 5.

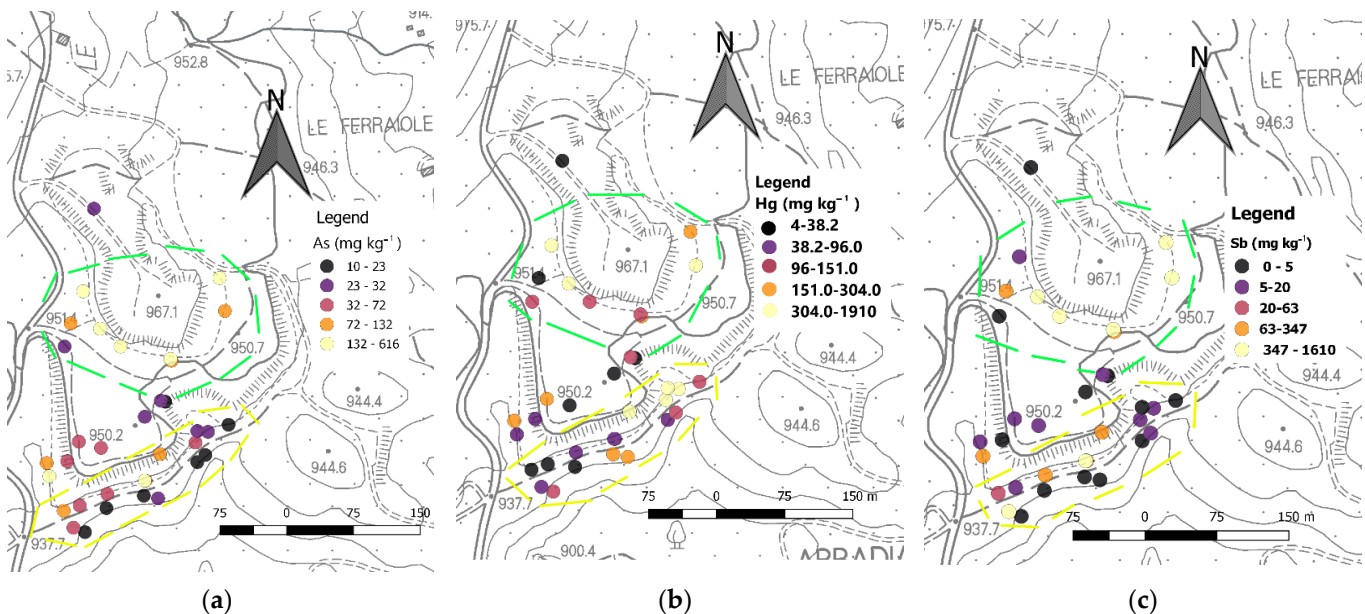

**Figure 5.** The dots maps of the studied topsoils of As (**a**), Hg (**b**) and Sb (**c**). The yellow and pale-green dashed lines are probably defining the areas where the post-roasting processes by the Gould and Cermak-Spirek furnace were located, respectively.

Average concentrations in uncontaminated soils of As, Hg, and Sb varied from 5 [51] 0.067 [52] to 0.3–8.6 mg kg$^{-1}$ [40,53], respectively. When comparing the concentrations of As, Hg, and Sb in Le Lame topsoils, it is observed that the high variability of concentrations found in these soils cannot only be attributed to soils developed on parent rock. The concentrations of As and Sb in this study are different with respect to those found by [1]. These authors reported concentrations of As and Sb of 77.5 and 138 mg kg$^{-1}$, respectively, in the lower part of the mining area and low concentrations in the upper part. Conversely, in this work, the highest concentrations of these two elements were found in the upper part of the mining dump. This difference can likely be related to both the low number of samples analyzed by [1] and the high heterogeneous distribution of As, Hg and Sb in the Le Lame topsoils, as shown in this work. Contamination by As and Sb is relatively common in soils within and around As and Sb mining sites where contents up to 10,250 [54–57] and 9619 mg kg$^{-1}$ [57–59], respectively, were recorded.

The concentration of Hg obtained in this study is in agreement with those determined by [1,25] in mining waste soils from the former-mining site of Abbadia San Salvatore. As previously reported, the high concentrations of Hg within mining waste soils are likely due to the incomplete thermal destabilization of cinnabar during roasting (unconverted cinnabar) and formation of insoluble Hg compounds during both ore processing and chemical weathering ([26], and references therein; [1]). Whatever the furnace used, the aim was to extract as much mercury as possible, although the Cermak-Spirek furnaces were far less efficient than the Gould ones. This may explain why, despite the fact that the ore deposit was enriched in Hg, the highest contents as well as the mean concentrations were recorded for As (1910 and 235 mg kg$^{-1}$, respectively) and Sb (1610 and 237.7 mg kg$^{-1}$, respectively). The leaching tests seem to confirm this consideration as the concentrations of Hg were on average lower than those of As and Sb (Table 6).

### 4.2. Leachates and PHREEQ-C Simulations

As previously mentioned, to simulate the effects deriving by soil-meteoric water interactions and to assess the behavior of As, Hg and Sb in solution, the topsoils were leached with $CO_2$-saturated MilliQ water. Due to interaction between $CO_2$ and MilliQ

water (2), $HCO_3$ in the eluates (Table 3) were clearly overestimated [32] with respect to the real concentrations:

$$CO_2 + H_2O = HCO_3^- + H^+ \tag{2}$$

The release of $H^+$ in solution produces a pH decrease down to 4.3, thus favoring the dissolution of carbonates and the removal of adsorbed metals on the surface of clay and other minerals. However, the alkaline hydroloysis proved to be effective, since all the post-leaching pH values were mostly circumneutral or slightly alkaline. It is clear that the leaching tests carried out in the present study affected the leachate chemistry. Nevertheless, the resulting aqueous solutions were characterized by low concentrations of Cl and Na, which is a typical feature of the surface and ground waters of the ASS area where the dominant species are in most cases $Ca (Mg)^{2+}$ and $SO_4^{2-}$ (e.g., [27]). It is worth noting that a significant positive correlation (0.82) occurs between $K^+$ and $NO_3^-$, suggesting the presence of an anthropogenic contamination ([60] and references therein), as also suggested by the numerous waste materials that are uncontrolledly left in the Le Lame mining dump area.

### 4.2.1. As in Eluates

Arsenic concentrations in the leachates (Table 6) have highlighted that 19 samples out of 35 exceeded the legal limit (10 $\mu g \cdot L^{-1}$) imposed by the Italian Legislative Decree 152/2006 for drinking waters. From PHREEQC simulations, As in solutions was occurring as $As^{5+}$, and the three main species were $HAsO_4^{2-}$, $H_2AsO_4^-$ e $CaAsO_4^-$. Additionally, PHREEQC simulations highlighted that the solubility of Fe-oxy-hydroxides (HFO), which could adsorb As and Sb on their surfaces, is apparently governed by the solubility of magnetite (SI = c.a.10, Table S1). This is in agreement with [61,62], who found that the retention of Arsenic at neutral condition pH (6–7) seems to be governed by HFO adsorption (e.g., [63,64]). At pH > 8, calcite may be precipitating. Thus, As can also enter the $CaCO_3$ lattice as calcium arsenate ($CaAsO_4$) [65,66], where Fe(Mn)-oxides lose their adsorbing efficiency [67]. The Eh-pH diagrams (Figure 6), representing the interaction between As-S-Ca, were constructed by taking the average As concentrations in the various eluates.

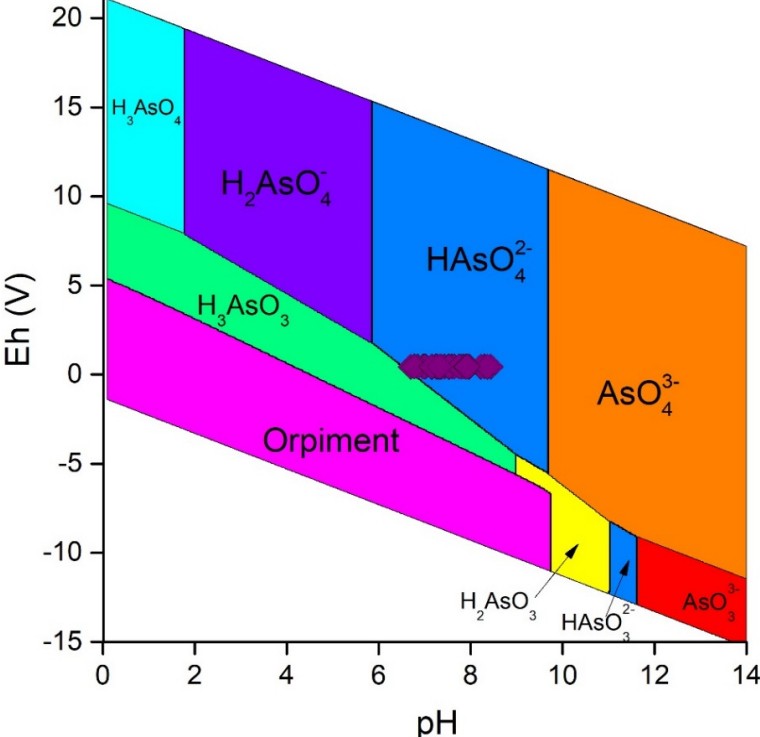

**Figure 6.** Eh–pH diagram of As–S. Purple diamonds represent the $CO_2$-saturated eluates.

Figure 6 confirms that the main As species in solution was $HAsO_4^{2-}$, although for samples with weakly acidic pH, a slight change in redox conditions can result in the reduction of $As^{5+}$ to $As^{3+}$, which occurs in the neutral form of $H_3AsO_3^0$. According to [61], this molecule is very poorly adsorbed, and the reduction of $As^{5+}$ to $As^{3+}$ may be the main mechanism by which As is released into groundwaters, thus favoring slightly anaerobic conditions. This would also explain why, for example, the waters draining the Galleria XXII, which is located about 1 km south of Le Lame, showed As concentrations between 10 and 12 $\mu g \cdot L^{-1}$ ([68], and references therein).

### 4.2.2. Hg in Eluates

Mercury concentrations in the leachates, except for sample LM23 (7.7 $\mu g \cdot L^{-1}$), (Table 6) are lower than the maximum permissible concentration (1 $\mu g \cdot L^{-1}$) for drinkable water. However, considering the distribution of the data, the sample LM23 can be considered as an outlier. According to the Eh-pH diagram of Hg-Cl-S (Figure 7) and the PHREEQC simulations, the main Hg species present in all samples is $Hg^0$. This is apparently confirmed by [1], who stated that about 60% of total Hg at the Le Lame mining dump is $Hg^0$. Nevertheless, a decrease in pH allows Hg to enter the stability range of HgS. This can be attributed to the leachate acidity (ca. 4.3), which favors the solubilisation of Me-sulfides occurring in soils and consequent re-precipitation of metals, which decreases their content in solution. The low concentrations of Hg in the leachates can be related to either the high efficiency of the roasting process or because Hg is not retained by HFO (e.g., [69,70]).

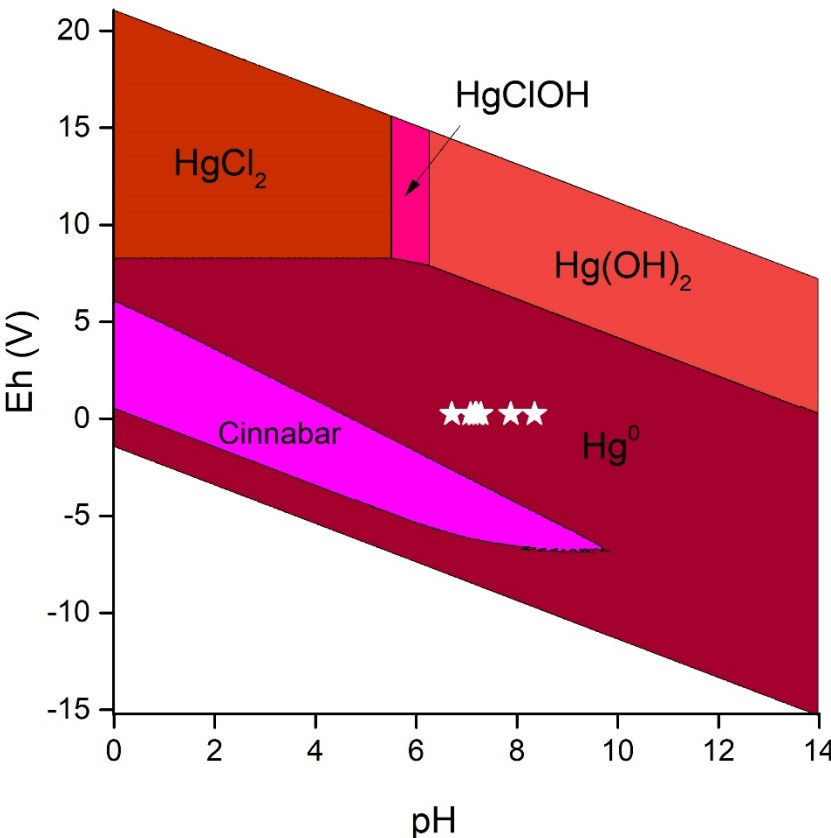

**Figure 7.** Eh–pH diagrams of Hg–S–Cl. White stars represent the seven $CO_2$-saturated eluates with concentrations >0.1 $\mu g \cdot L^{-1}$.

### 4.2.3. Sb in Eluates

Due to the strong bond with Fe-oxy-hydroxides and the electronic configuration similar to that of As, Sb is generally considered to be relatively immobile. According to [71], the transformation of mobile forms of Sb is predominantly controlled by naturally

occurring precipitation and adsorption processes (e.g., [72]). Sixteen samples out of 35 (Table 6) showed concentrations above the legal limit of Sb (5 $\mu g \cdot L^{-1}$) for drinking water. PHREEQC simulations revealed that Sb in solutions is mostly dominated by $Sb^{5+}$ and the main three species were $SbO_3^-$, $Sb(OH)_6^-$ e $CaSb(OH)_6^+$. In the Eh-pH diagram of Sb-S-Ca-Fe (Figure 8), all $CO_2$-saturated leachates are included in the stability range of $SbO_2$ (usually known as cervantite- $Sb_2O_4$). Despite the fact that cervantite was not found is the ASS mine, it was recognized as the primary mineral in several Sb-mines from Tuscany (e.g., the Selva mine and the Le Cetine mine ([73–75] and reference therein)). However, all Sb-compounds are also found close to the stability range of tripuhyte ($FeSbO_4$) and romeite ($Ca_2Sb_2O_7$). As well as for As, $Sb(OH)_6^-$ and $SbO_3^-$ (two of the predominant water-soluble Sb compounds at neutral pH), can be co-precipitating with $Ca^{2+}$ ions to produce insoluble and stable calcium antimonate $Ca[Sb(OH)_6]_2$ ([76], and references therein) or romeite that may form over calcite crystals.

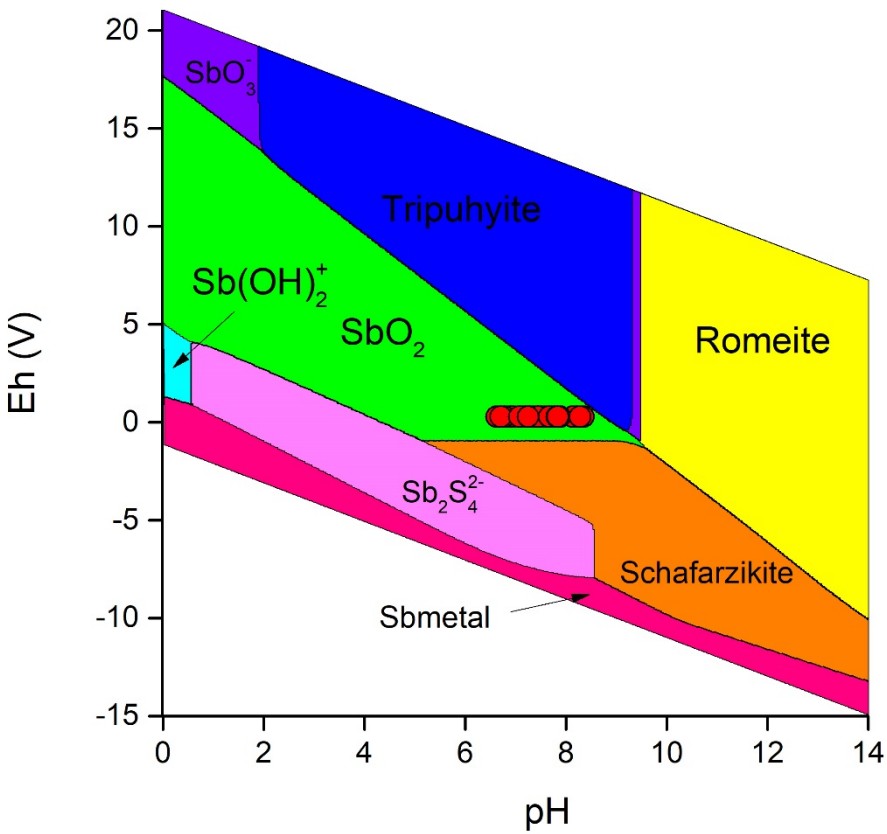

**Figure 8.** Eh–pH diagrams of Sb–S–Ca–Fe. Red circles represent the $CO_2$-saturated eluates.

As previously evidenced, PHREEQC simulations indicate that tripuhyte was always occurring, with the exception of sample LM31. Tripuhyte is not found as the main mineral because it is likely that part of hosted-$Fe^{2+}$ tends to precipitate as Fe-oxy-hydroxide. Nevertheless, its presence was confirmed by various PHREEQC simulations, which also showed that SI always clustered around 1.5 (Table S1). The thermodynamic analyses (log K, Table 2) of tripuhyte demonstrated that this mineral phase is stable and is recognized as crucial sink of Sb in the natural environment [42]. Therefore, it is clear that the formation of such stable secondary Sb minerals (romeite and tripuhyte) via precipitation plays a key role in the sequestration of mobile species of Sb in the natural environment [42,71,77]. To confirm the presence of tripuhyte in soils, although its value is expected to be low, a detailed mineralogical study of the topsoils is required.

## 5. Conclusions

To be best of our knowledge, this study provided the very first detailed investigation on the distribution of As, Hg, and Sb (total and leachate) in the topsoils from the Lame dumping area. Here, the roasting products from the former Hg-mining area of Abbadia San Salvatore (Mt. Amiata district, central Italy), resulting after the extraction of Hg, were accumulated. Relatively high contents of the three elements were determined in the soil fraction as well as in the leachate; the latter analyzed after reacting $CO_2$-saturated MilliQ water with powdered soils in a 5:1 ratio to simulate the effects deriving from the interaction between soils and meteoric waters (although the concentrations of Hg in the leachates were on average lower than those recorded for As and Sb). Nevertheless, differences in the distribution of Hg were observed since the sites where the "*rosticci*" derived by the older and less efficient Cermak-Spirek furnaces showed much higher concentrations when compared to those produced by the more recent and productive Gould furnaces. Simulations by the PHREEQC geochemical code on the leachate samples allowed the definition of the fate of As, Hg, and Sb, the last of which was more mobile. The speciation of the various elements in aqueous solutions is critical, since the oxidation state can determine the degree of toxicity to living organisms. For example, e.g., $As^{3+,}$ and $Sb^{3+}$ are known to be more toxic than $As^{5+}$ and $Sb^{5+}$, respectively [78,79]. Consequently, this study can also be important in evaluating the effects produced by the soil-waste-meteoric interaction processes since the three elements can be transferred with a different rate to the local groundwater system. Consequently, detailed investigations on springs and groundwaters are to be investigated although, as already mentioned, previous studies have not highlighted critical concentrations of As and Sb in the waters downstream from the study area.

**Supplementary Materials:** The following are available online at https://www.mdpi.com/article/10.3390/app11177893/s1, Table S1: Saturation index for the main minerals computed by the PHREEQC geochemical code.

**Author Contributions:** Conceptualization, F.M., G.M. and O.V.; methodology, F.M., B.N. and O.V.; software, F.M. and G.M.; validation, F.M., G.M. and O.V.; formal analysis, F.M., B.N., O.V., M.L.; investigation, F.M., G.M., O.V., B.N., M.L.; resources, O.V. and D.R.; data curation, F.M., B.N., O.V., M.L.; writing—original draft preparation, F.M., G.M. and O.V.; writing—review and editing, F.M., G.M., D.R., B.N., M.L. and O.V.; funding acquisition, O.V. and D.R. All authors have read and agreed to the published version of the manuscript.

**Funding:** This research received no external funding.

**Acknowledgments:** Many thanks are due to A. Esposito, M. Niccolini, F. Bianchi and F. Piccinelli for their help during the sampling sessions. The Authors are in debt with the Editor and two anonymous reviewers whose comments and suggestions improved an early version of the manuscript.

**Conflicts of Interest:** The authors declare no conflict of interest.

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
