# Peer review of "Total and Leached Arsenic, Mercury and Antimony in the Mining Waste Dumping Area of Abbadia San Salvatore (Mt. Amiata, Central Italy)"

_applsci, doi:10.3390/app11177893_

Round 1
Reviewer 1 Report
The manuscript presents interesting results. The problem of leaching the arsenic, mercury and antimony from mining landfills is very important for the environmental protection. The authors try to determine the total amount of Hg, As and Sb in waste top-soils in the waste dumping area of Abbadia San Salvatore (Italy) and to quantify the release of Hg, As, Sb from leaching tests, carried out with CO2-saturated Milli-Q water. They also try to assess the speciation of As, Hg, Sb and define whether the three chalcophile elements were preferentially partitioned in solution or were coprecipitating with other minerals by simulations carried out by the PHREEQC geochemical code.
However, I have some comments on this manuscript. Some things need to be improved.
- The Study area section should be supplemented. It should be described in which part of the studied area the waste from which furnace was stored. Mark this information in Figure 2.
- The Materials and Methods section needs to be supplemented:
- It is not clear how the samples were stored or how they were transported. Were they averaged prior to analysis? What acids were used to dissolve the samples?
- No information on the accuracy of the determinations.
- There is no information on the statistical tests using to analyse results (this information is in the Discussion section but should be describe in methodology section.
- What statistical software was used to analyse the data?
- Figure 2 should be supplemented with the sample numbers
- Lines 198-200 - this information should be placed in the Materials and methods section.
- line 200 – min content of Sb = 0.18 mg kg-1 ? In Table 2 we can find “<2”. It is not clear.
- Table 2 - Sometimes the results are given to two decimal places and sometimes to one. This should be harmonized.
- Figure 5 - the boundaries of the landfill of waste from different furnaces should be marked.
- The Conclusion section contains information that was not discussed in the discussion, see line 439-440.
Reviewer 2 Report
The manuscript evaluates distribution and concentration of As, Hg, and Sb in top-soil samples of a landfill, which tailing of Hg-bearing minerals disposed there. The manuscript addressed a critical issue in and around landfills. The authors successfully define the As, Hg, and Sb content in the study area and showing the urgent need of action in terms of As and Sb pollution in the site. The findings of this study will improve the status of pollution control regarding mining tailings and will act as a reference for other studies.
The manuscript has some issues that I listed below:
Major points
- The related literature review for the issue is poorly developed.
- I highly recommend that the authors try to add a brief local geological feature in order to justify the high content of Ca and Mg in the leached samples.
And also, develop the role of Fe-oxides and hydroxides to limit the As, Sb, and Hg (in particular) mobility. - Totally speaking, I do believe that the authors should talk about the environmental issues of As and Sb pollution in the site and mention the potential health risk to nearby ecosystems and human communities.
Minor points
- In-text reference missing: 46–53-line number in page 2 (as a rule when you use other works should get credit to them).
- I recommend that all detail regarding mining area bring in the Study Area sub-section.
- In Figure 1, the map did not have a scale, it needs to show the readers the extent of study area and each section.
- Formula in page 3 does have a number.
- In Figure 2, Tuscany and Abbadia map is not clear.
- Table 5 is more fitted into the result section.
